**Data Availability Statement:** All relevant data are within the manuscript and its Supporting information files.

**Funding:** This work is partially supported by the Guangdong Province Ordinary Universities Key

# Research on reachable set boundary of neutral system with various types of disturbances

**Dongmei Xia**[1]*, **Kaiyuan Chen**[1], **Lin Sun**[2]

1 Foshan Polytechnic, Foshan, Guangdong Province, China, 2 Shenzhen Yingshisheng Information Technology Co., Ltd., Shenzhen, Guangdong, China

\* dreamchaser_edu@163.com

## Abstract

This study delves into neutral-type systems (NTSs), emphasizing the critical role of defining precise reachable set (RS) boundaries for safe and efficient system design and operation. The investigation notably addresses the challenges posed by time-varying delays, applying Lyapunov's direct method alongside advanced matrix inequality techniques to identify minimized and more accurate ellipsoidal boundaries of the RS in NTSs influenced by bounded and nonlinear disturbances. Our findings, verified through numerical simulations and comparisons with existing literature, demonstrate enhanced control and management capabilities for complex systems, thus underscoring the substantial theoretical and practical value of incorporating delay elements in NTSs.

## 1 Introduction

Neutral-type systems (NTSs) are a class of dynamic systems characterized by unique delay attributes, where the differential equations of the system incorporate not only delays in the state but also in the derivatives of the state [1–3]. This distinct structure allows NTSs to more accurately describe the dynamic behaviors of certain physical, engineering, and biological systems, especially excelling in handling issues like transmission delays and control execution delays [4–6]. The study of NTSs is crucial for enhancing the prediction, stability analysis, and design of control strategies for complex systems. NTSs are extensively used in fields such as automatic control, signal processing, and network communications, helping to improve the safety and efficiency of systems.

Research on NTSs can be traced back to the early 20th century, but more systematic theoretical research began in the 1960s. With the development of control theory, especially in the study of delay effects in dynamical and complex systems, the importance of NTSs has gradually been recognized. The reachable set (RS) is a crucial concept in control theory, describing the collection of all possible endpoint states that a system can achieve given any initial state and possible control inputs [7–11]. In summary, studying the RS of NTSs can help assess the stability and safety of the system when facing various external disturbances and internal changes.

Areas Special Project (High-end Equipment
Manufacturing), Project Number: 2022ZDZX3074.
The funders contributed to the data collection and
analysis.

**Competing interests:** The authors have declared
that no competing interests exist.

Understanding the states that a system might reach under specific conditions is crucial for preventing potential risks and implementing necessary safety measures.

In the study cited as [11], Kim focuses on identifying an ellipsoidal RS that captures the state boundaries of linear systems experiencing time delays and affected by disturbances with bounded peaks. In [12], Lam explores the calculation of RS for discrete-time linear systems characterized by multiple constant delays and bounded peak inputs. By employing a newly formulated Lyapunov-Krasovskii functional (LKF), Lam successfully establishes delay-dependent conditions that facilitate the resolution of this estimation challenge. The research results include methods for enclosing the system's RS with ellipsoids and validating the effectiveness of these methods through numerical examples. Furthermore, these findings have been extended to systems with polyhedral parameter uncertainties. In [13], by employing the Lyapunov-Razumikhin method and solving a series of linear matrix inequalities related to the upper bounds of delay lengths, a successful definition of an ellipsoid was achieved. This ellipsoid defines the Euclidean space of the state set, which is reachable from the origin within a finite time via inputs with peak constraints. In [14], Lee and Kim presented an advanced method for estimating RS in linear systems with time-varying delays by leveraging the LKF methodology. This approach incorporates the use of integral inequalities and enhanced zero equality techniques, utilizing linear matrix inequalities (LMIs) to precisely evaluate the system's dynamic behavior, and the reduced conservatism of this approach was validated through numerical examples. Different from other research work on RS [15–17], Zhang and Lam developed a state feedback controller design and RS estimation method for linear systems with distributed time delay and bounded disturbance input by applying LKF and delay partition technology, aiming at ensuring that the closed-loop system state is effectively controlled within the preset ellipsoid range, and its effectiveness was verified by numerical examples and simulations [18]. In [19], Kwon and Lee considered more general time derivative conditions and constructed a new method for solving LMIs using the Lyapunov method to establish the RS of the system. Furthermore, they validated its effectiveness through numerical simulations by comparison. In the study cited as [20], the focus is on examining the boundedness of the RS in linear discrete-time systems that experience state delays and are impacted by bounded disturbances. This innovative approach goes beyond merely shrinking their radius; it strategically minimizes the projection distance of ellipsoids, applying different exponential convergence rates along each axis, thus achieving a smaller boundary. As a result, the convergence of these ellipsoids creates a tighter boundary for the RS.

The investigation of the RS for NTSs presents more complex and challenging scenarios than typical time-delay systems, both in theoretical exploration and practical application. In areas such as control engineering and signal processing, predicting and managing the behavior of NTSs is notably more challenging than dealing with systems lacking delays in their derivatives [21–25]. In [26], Shen and Wang have developed a data-driven control approach for networked nonlinear systems with event-triggered outputs, using an advanced observer to estimate unknown disturbances. They defined triggering conditions for both single and multi-input/output systems based on estimated and actual tracking errors. Sufficient conditions were established to ensure the system's tracking errors remain ultimately bounded, with numerical examples confirming the strategy's effectiveness. In [27], Qin and colleagues developed an adaptive safety tracking control strategy suitable for nonlinear systems with both matched and mismatched uncertainties. This strategy optimizes system performance through control barrier functions (CBF) and event-triggering mechanisms, uses adaptive neural networks to precisely approximate solutions, and employs Lyapunov theory to ensure the uniform ultimate boundedness (UUB) of system errors. Simulation results validate the effectiveness of this method. In [28], Jiang and Xia investigated the RS estimation for a specific category of

Markov jump neutral neural networks, which are influenced by bounded disturbances and exhibit time-varying delays. With the help of delay segmentation method, numerical simulation confirmed the correctness of theoretical analysis. Similarly intriguing as the work in Reference [27], Qin and Qiao et al. have developed an adaptive stabilization strategy based on the CBF for nonzero-sum differential games in uncertain nonlinear systems with state constraints. This strategy takes into account random disturbances and uncertainties in the control input matrix, utilizing a nominal system and custom cost functions to effectively transform the robust control problem of multiple players into an optimal control problem. By employing Lyapunov theory, it ensures that the system states and neural network weights are uniformly UUB, with the effectiveness of the approach validated through two simulation examples [29].

In [30], Shen designed a new LKF, a LMI form of nonconvex scalar, which can effectively find the ellipsoid boundary of the RS of NTSs with bounded disturbances. Numerical comparison and simulation further show that this method can get a smaller RS. In [31], in order to find a smaller RS of neutral Markov jump system with disturbance, a new type of LMI is constructed based on Lyapunov function method. Although the research on RS is rarely put forward in neutral Markov jump system, this method can find a more suitable and smaller RS. In addition, an improved LKF is designed to solve the RS ellipsoid of neutral semi-Markov jump systems with disturbances [32]. Shen and Zhang conducted research on anti-disturbance control for Markov jump systems with matched and mismatched disturbances. They developed a composite static output control strategy that includes $H_\infty$ performance to ensure system stability. Additionally, they introduced an integral sliding-mode output (ISMO) control strategy that effectively mitigates disturbances using their upper bounds. The effectiveness of their research was demonstrated through numerical simulations [33]. Different from [17, 34], Zhang investigated how to estimate the RS for Markov jump bidirectional associative memory neural networks experiencing time-varying delays and bounded disturbances in their inputs. This research aims to understand how these factors influence the network's behavior and stability. Numerical examples show that the simulation results are effective. In [35], Qin and Zhu developed a new dynamic event-triggered safety control strategy based on integral reinforcement learning, specifically designed to address multi-player Stackelberg-Nash game problems in continuous-time nonlinear systems with time-varying state constraints. They simplified the constrained problem by integrating innovative barrier functions with state transformation techniques, and implemented a dynamic event-triggering mechanism along with a single critic neural network to ensure system stability and effective control of errors.

Building on the insights from the aforementioned literature and our investigation into the RS analysis of NTSs, it is evident that analyzing the RS of NTSs is significantly more challenging compared to conventional dynamic systems. This complexity arises from the influence of delay elements, which add layers of difficulty to the theoretical study [25, 36]. Time delays not only impact system stability but can also fundamentally alter system behavior, potentially causing oscillations or instability [37, 38]. In light of these challenges, this paper advances existing research by employing matrix inequality techniques and proposing a novel Lyapunov functional method to identify smaller, more precise RS boundaries for NTSs. The primary contributions of this work are summarized as follows.

1. Firstly, many practical engineering systems inherently exhibit delay characteristics, making the study of neutral-type systems (NTSs) essential. This research is crucial for designing stable and robust control systems, as it provides theoretical insights and methodological solutions to effectively tackle the challenges associated with delays. This aspect is particularly significant for real-world applications where delay elements can critically affect system performance.

2. Furthermore, this work delves into the determination of elliptical boundaries for the RS in NTSs. By developing innovative Lyapunov functionals and leveraging advanced matrix inequality techniques, several methodologies were proposed to identify minimal elliptical boundaries of the RS. These contributions enhance both the theoretical foundation and practical applicability of RS analysis in NTSs, which are essential for progress in control design, system optimization, and safety assurance.

3. Finally, compared to other research efforts on the RS of NTSs (e.g., [39, 40]), numerical simulations using MATLAB/SIMULINK have demonstrated that the proposed approach not only effectively encompasses all system trajectories but also achieves smaller boundaries for the ellipsoidal RS, underscoring its efficiency and accuracy.

The rest of this paper is organized as follows: Section 2 provides a detailed introduction to the model framework of NTSs, including some key lemmas and the control objectives of the study. In the Section 3, we propose a new method to determine the ellipsoidal boundary of the RS for NTSs using Lyapunov functionals. The section 4 validates the effectiveness of our method through comparative simulations with existing research. Finally, in the section 5, we summarize the main conclusions of this study and look forward to future research directions.

## 2 Model description and useful lemma

### 2.1 Neutral type systems, NTSs

**Remark 1:** A neutral type system (NTS), also known as a neutral delay system or a delay-differential equation with derivative delays, characterizes a dynamic system where the state of a variable at any given time is affected by its current and historical values. Essentially, this system incorporates delays in both the state variables and their derivatives, reflecting a more complex dependency on past behavior [41–43].

Consider the general representation of a NTS, characterized by the following delay differential equation:

$$\dot{x}(t) = f(x(t), x(t - \tau), \dot{x}(t - \tau)) \tag{1}$$

where, $x(t)$ denotes the system's state at time $t$, $\dot{x}(t)$ indicates the rate of change of the state variable over time, $\tau$ represents the delay, and $f$ is a function that dictates the system's evolution through time.

From the analysis discussed, the NTS examined in this section can be characterized as

$$\begin{cases} \dot{x}(t) - C\dot{x}(t - \tau) = A_1 x(t) + A_2 x(t - h(t)) + Bw(t) \\ \qquad\qquad\qquad + D_1 f_1(t, x(t)) + D_2 f_2(t, x(t - h(t))), \\ x(t_0 + \theta) = \varphi(\theta), \ \forall \theta \in [-\tau^*, 0]. \end{cases} \tag{2}$$

where, $x(t) \in \mathbb{R}^n$ signifies the state vector of the system, $w(t) \in \mathbb{R}^l$ corresponds to disturbances, and $\varphi(\cdot)$ is a differentiable function defining initial conditions. Here, $\tau$ is a constant delay used specifically in the term $\dot{x}(t - \tau)$, and $h(t)$ is a time-varying delay affecting the state term $x(t - h(t))$ and adds additional dynamism to the system's response to past states. Both the

neutral-type delay and the disturbances are subject to the following conditions:

$$0 \leqslant h(t) \leqslant h \leqslant +\infty \tag{3}$$

$$\dot{h}(t) \leqslant h_d \leqslant 1 \tag{4}$$

$$w^{\mathrm{T}}(t)w(t) \leqslant 1 \tag{5}$$

where, $h$ and $h_D$ are both constants, and $\tau^* = \max(\tau, h)$, $A \in \mathbb{R}^{n \times n}$, $B \in \mathbb{R}^{n \times n}$, $C \in \mathbb{R}^{n \times n}$, and $D \in \mathbb{R}^{n \times x}$ are known real matrices. Given positive numbers $\lambda_1$ and $\lambda_2$, nonlinear disturbances $\|f_1(t, x(t))\|$ and $\|f_2(t, x(t-h(t)))\|$ satisfy the conditions:

$$\| f_1(t, x(t)) \| \leqslant \lambda_1 \| x(t) \| \tag{6}$$

$$\| f_2(t, x(t-h(t))) \| \leqslant \lambda_1 \| x(t-h(t)) \| \tag{7}$$

Then, we define the RS of the system Eq (2) as a set satisfying the following conditions:

$$\mathfrak{R}_x = \{x(t) \in \mathbb{R}^n \mid x(t), w(t) \text{ satisfies conditions } Eq.(2) - Eq.(7)\} \tag{8}$$

Next, we will use the ellipsoidal boundary $\mathfrak{I}(P_1, 1)$ defined below to represent the $\mathfrak{R}_x$:

$$\mathfrak{I}(P_1, 1) = \{x(t) \in R^n \mid x(t) \text{ satisfies } Eq.(2) - Eq.(7), \text{ and } x^{\mathrm{T}}(t)P_1 x(t) \leqslant 1\} \tag{9}$$

**Remark 2:** The matrices $A$, $B$, $C$, and $D$ are known real matrices used to describe the interactions within the system and between the system and external disturbances. To analytically describe the reachability set of the system, an ellipsoidal boundary $\mathfrak{I}(P_1, 1)$ is defined, within which all state trajectories of the system must be contained. It includes all states $x(t)$ that not only comply with the system's differential equations but also meet the quadratic constraints defined by the matrix $P_1$.

## 2.2 Main useful lemmas

**Definition 1:** A system can possess no equilibrium points, one equilibrium point, or several. Nonlinear systems typically exhibit more complex behaviors at their equilibrium points, which can include having multiple equilibrium points or none at all. In contrast, for linear systems represented by

$$\dot{x} = Ax, \tag{10}$$

When $A$ is invertible, the system uniquely stabilizes at the equilibrium point $x = 0$.

**Definition 2** ([44]): If there are constants $\alpha > 0$ and $\gamma \geq 1$ such that the inequality below holds for every $x(t)$:

$$\| x(t) \| \leqslant \gamma \sup_{-\gamma^* < s < 0} \sqrt{\| \varphi(s) \|^2 + \| \phi(s) \|^2} \mathrm{e}^{-\alpha t} \tag{11}$$

Then, system Eq (1) is deemed exponentially stable, characterized by a decay rate determined by $\alpha$.

**Lemma 1** ([45]): For specified symmetric positive definite matrices $\Sigma_1$, $\Sigma_2 > 0$ and any matrix $\Sigma_3$, the condition $\Sigma_1 + \Sigma_3^{\mathrm{T}} \Sigma_2^{-1} \Sigma_3 < 0$ is met if both of the following matrix inequalities

are true:

$$\begin{bmatrix} \Sigma_1 & \Sigma_3^{\mathrm{T}} \\ \Sigma_3 & -\Sigma_2 \end{bmatrix} < 0, \quad \begin{bmatrix} -\Sigma_2 & \Sigma_3 \\ \Sigma_3^{\mathrm{T}} & \Sigma_1 \end{bmatrix} < 0. \tag{12}$$

**Lemma 2** ([46]): Given any matrix $\Phi$ in $\mathbb{R}^{n \times n}$ and a constant $\gamma > 0$, if the function $w : [0, \gamma] \to \mathbb{R}^n$ can be integrated, then the equation below is satisfied:

$$\left( \int_0^\gamma w(s)\mathrm{d}s \right)^{\mathrm{T}} \Phi \left( \int_0^\gamma w(s)\mathrm{d}s \right) \leqslant \gamma \int_0^\gamma w^{\mathrm{T}}(s)\Phi w(s)\mathrm{d}s \tag{13}$$

**Lemma 3** ([47]): Let $V(x(0)) = 0$ and $w^{\mathrm{T}}(t)w(t) \leqslant w_m^2$. If the following condition is satisfied:

$$\dot{V}(t, x_t) + \alpha V(t, x_t) - \beta w^{\mathrm{T}}(t)w(t) \leqslant 0 \tag{14}$$

then, for all $t \geqslant t_0$, it holds that

$$V(t, x_t) \leqslant \frac{\beta}{\alpha} w_m^2 \tag{15}$$

**Lemma 4** ([48]): Assuming $\alpha \in \mathbb{R}^{n_a}$, $\beta \in \mathbb{R}^{n_b}$, and $N \in \mathbb{R}^{n_a \times n_b}$, when the block matrix $\begin{bmatrix} X & Y \\ Y^{\mathrm{T}} & Z \end{bmatrix} \geqslant 0$ holds, then for any matrices $X \in \mathbb{R}^{n_a \times n_a}$, $Y \in \mathbb{R}^{n_a \times n_b}$, and $Z \in \mathbb{R}^{n_b \times n_b}$, the subsequent inequality is satisfied:

$$-2\alpha^{\mathrm{T}}N\beta \leqslant \begin{bmatrix} \alpha \\ \beta \end{bmatrix}^{\mathrm{T}} \begin{bmatrix} X & Y - N \\ Y^{\mathrm{T}} - N^{\mathrm{T}} & Z \end{bmatrix} \begin{bmatrix} \alpha \\ \beta \end{bmatrix} \tag{16}$$

**Lemma 5** ([49]): For a vector function $x(t)$ belonging to $\mathbb{R}^n$ that is both continuous and first-order differentiable, consider any matrix $F = \begin{bmatrix} F_1 & F_2 & F_3 & F_4 & F_5 & F_6 & F_7 & F_8 \end{bmatrix}$ and a positive scalar $Z_1 > 0$. Additionally, assume there is a continuous function $h(t)$ where $h$ is a positive real constant. Under these conditions, the following inequality is satisfied when $0 \leqslant h(t) \leqslant h$:

$$-\int_{t-h(t)}^{t} \dot{x}^{\mathrm{T}}(\alpha)Z_1\dot{x}(\alpha)\mathrm{d}\alpha \leqslant \xi^{\mathrm{T}}(t)\Xi_1\xi(t) + h\xi^{\mathrm{T}}(t)X^{\mathrm{T}}Z_1^{-1}F\xi(t) \tag{17}$$

where

$$\xi(t) = \begin{bmatrix} x^{\mathrm{T}}(t) & \dot{x}^{\mathrm{T}}(t) & x^{\mathrm{T}}(t - h(t)) & x^{\mathrm{T}}(t - h) & \dot{x}^{\mathrm{T}}(t - \tau(t)) & w^{\mathrm{T}}(\sigma(t)) & f_1^{\mathrm{T}} & f_2^{\mathrm{T}} \end{bmatrix}^{\mathrm{T}} \tag{18}$$

$$\Xi_1 = \begin{bmatrix} F_1 + F_1^{\mathrm{T}} & F_2 & F_3 - F_1^{\mathrm{T}} & F_4 & F_5 & F_6 & F_7 & F_8 \\ * & 0 & -F_2^{\mathrm{T}} & 0 & 0 & 0 & 0 & 0 \\ * & * & -F_3 - F_3^{\mathrm{T}} & -F_4 & -F_5 & -F_6 & -F_7 & -F_8 \\ * & * & * & 0 & 0 & 0 & 0 & 0 \\ * & * & * & * & 0 & 0 & 0 & 0 \\ * & * & * & * & * & 0 & 0 & 0 \\ * & * & * & * & * & * & 0 & 0 \\ * & * & * & * & * & * & * & 0 \end{bmatrix}. \tag{19}$$

Applying a technique analogous to that used in Lemma 5 enables us to derive Lemma 6 and Lemma 7.

**Lemma 6** ([30]): For a vector function $x(t) \in \mathbb{R}^n$ that is continuous and differentiable, given any matrix $X = \begin{bmatrix} X_1 & X_2 & X_3 & X_4 & X_5 & X_6 & X_7 & X_8 \end{bmatrix}$ and a positive scalar $Z_2 > 0$, along with a continuous function $h(t)$ where $h > 0$, the ensuing inequality is valid provided $0 \leqslant h(t) \leqslant h$:

$$-\int_{t-h}^{t-h(t)} \dot{x}^{\mathrm{T}}(\alpha) Z_2 \dot{x}(\alpha) \mathrm{d}\alpha \leqslant \xi^{\mathrm{T}}(t) \Xi_2 \xi(t) + h \xi^{\mathrm{T}}(t) X^T Z_2^{-1} F \xi(t) \tag{20}$$

where

$$\xi(t) = \begin{bmatrix} x^{\mathrm{T}}(t) & \dot{x}^{\mathrm{T}}(t) & x^{\mathrm{T}}(t-h(t)) & x^{\mathrm{T}}(t-h) & \dot{x}^{\mathrm{T}}(t-\tau(t)) & w^{\mathrm{T}}(\sigma(t)) & f_1^{\mathrm{T}} & f_2^{\mathrm{T}} \end{bmatrix}^{\mathrm{T}} \tag{21}$$

$$\Xi_2 = \begin{bmatrix} 0 & 0 & X_1^{\mathrm{T}} & -X_1^{\mathrm{T}} & 0 & 0 & 0 & 0 \\ * & 0 & X_2^{\mathrm{T}} & -X_2^{\mathrm{T}} & 0 & 0 & 0 & 0 \\ * & * & X_3 + X_3^{\mathrm{T}} & X_4 - X_3^{\mathrm{T}} & X_5 & X_6 & X_7 & X_8 \\ * & * & * & X_4 - X_3^{\mathrm{T}} & -X_5 & -X_6 & -X_7 & -X_8 \\ * & * & * & * & 0 & 0 & 0 & 0 \\ * & * & * & * & * & 0 & 0 & 0 \\ * & * & * & * & * & * & 0 & 0 \\ * & * & * & * & * & * & * & 0 \end{bmatrix}. \tag{22}$$

**Lemma 7** ([50]): Assume $x(t) \in \mathbb{R}^n$ is a continuously differentiable vector function. Take any matrix $Y = \begin{bmatrix} Y_1 & Y_2 & Y_3 & Y_4 & Y_5 & Y_6 & Y_7 & Y_8 \end{bmatrix}$, with each $Y_i$ representing a sub-matrix or vector, and let $Z_3$ be a positive scalar greater than zero. Furthermore, suppose $h(t)$ is a continuous function satisfying $0 \leq h(t) \leq h$, where $h$ is a fixed positive real number. In this context, the following inequality is satisfied:

$$-\int_{t-h}^{t} \dot{x}^{\mathrm{T}}(\alpha) Z_3 \dot{x}(\alpha) \mathrm{d}\alpha \leqslant \xi^{\mathrm{T}}(t) \Xi_3 \xi(t) + h \xi^{\mathrm{T}}(t) X^T Z_3^{-1} F \xi(t) \tag{23}$$

where

$$\xi(t) = \begin{bmatrix} x^{\mathrm{T}}(t) & \dot{x}^{\mathrm{T}}(t) & x^{\mathrm{T}}(t-h(t)) & x^{\mathrm{T}}(t-h) & \dot{x}^{\mathrm{T}}(t-\tau(t)) & w^{\mathrm{T}}(\sigma(t)) & f_1^{\mathrm{T}} & f_2^{\mathrm{T}} \end{bmatrix}^{\mathrm{T}} \tag{24}$$

$$\Xi_3 = \begin{bmatrix} Y_1 + Y_1^{\mathrm{T}} & Y_2 & Y_3 & -Y_1^{\mathrm{T}} + Y_4 & Y_5 & Y_6 & Y_7 & Y_8 \\ * & 0 & 0 & -Y_2^{\mathrm{T}} & 0 & 0 & 0 & 0 \\ * & * & 0 & -Y_3^{\mathrm{T}} & 0 & 0 & 0 & 0 \\ * & * & * & -Y_4 - Y_4^{\mathrm{T}} & -Y_5 & -Y_6 & -Y_7 & -Y_8 \\ * & * & * & * & 0 & 0 & 0 & 0 \\ * & * & * & * & * & 0 & 0 & 0 \\ * & * & * & * & * & * & 0 & 0 \\ * & * & * & * & * & * & * & 0 \end{bmatrix} \tag{25}$$

## 2.3 Control objective

This paper aims to apply the direct Lyapunov method, the use of free weighting matrices, and matrix inequality techniques to identify a suitable Lyapunov function and establish matrix inequality forms that ascertain the minimal ellipsoidal boundary for NTSs. It is important to note that the method proposed is not optimal, nor does it yield the smallest ellipsoidal RS. Compared to existing research on RS for NTSs, our method is capable of identifying more compact RS. Through numerical examples, we illustrate that our proposed technique more precisely pinpoints smaller and more efficient RS for NTSs than the methods previously established.

## 3 Ellipsoidal reachable set of neutral system

In this section, we address the RS boundary issue for the aforementioned NTSs, choose the suitable Lyapunov functional, and present the following theorem:

**Theorem 1:** For the neutral system Eq (2), if there exist real matrices $X = [X_1, X_2, X_3, X_4, X_5, X_6, X_7, X_8]$, $Y = [Y_1, Y_2, Y_3, Y_4, Y_5, Y_6, Y_7, Y_8]$, and $Z = [Z_1, Z_2, Z_3, Z_4, Z_5, Z_6, Z_7, Z_8]$, along with the matrices $Q_{12}$, $P_2$, and $P_3$, and symmetric matrices $Q_{11}, Q_{22}, Q_{12}, P_1 > 0, R_1 > 0, R_2 > 0, R_3 > 0, S_1 \geqslant 0, S_2 > 0, S_3 > 0$, and $G > 0$, as well as a scalar $\alpha > 0$, such that the following linear matrix inequalities are satisfied:

$$\begin{bmatrix} \Phi & hF^{\mathrm{T}} & hX^{\mathrm{T}} & hY^{\mathrm{T}} \\ * & -\mathrm{e}^{-\alpha h}S_1 & 0 & 0 \\ * & * & -\mathrm{e}^{-\alpha h}S_2 & 0 \\ * & * & * & -\mathrm{e}^{-\alpha h}S_3 \end{bmatrix} \leqslant 0 \tag{26}$$

$$\begin{bmatrix} Q_{11} & Q_{12} \\ Q_{12}^{\mathrm{T}} & Q_{22} \end{bmatrix} \geqslant 0, \tag{27}$$

$$\begin{bmatrix} X & Y \\ Y^{\mathrm{T}} & Z + \dfrac{1}{\tau}\mathrm{e}^{-\alpha\tau}R \end{bmatrix} \geqslant 0 \tag{28}$$

where

$$\Phi = \begin{bmatrix} \Phi_{11} & \Phi_{12} & \Phi_{13} & \Phi_{14} & \Phi_{15} & \Phi_{16} & \Phi_{17} & \Phi_{18} \\ * & \Phi_{22} & \Phi_{23} & -hX_2^{\mathrm{T}} - hY_2^{\mathrm{T}} & P_3^{\mathrm{T}}C & P_3^{\mathrm{T}}B & P_3^{\mathrm{T}}D_1 & P_3^{\mathrm{T}}D_2 \\ * & * & \Phi_{33} & \Phi_{34} & \Phi_{35} & \Phi_{36} & \Phi_{37} & \Phi_{38} \\ * & * & * & \Phi_{44} & \Phi_{45} & \Phi_{46} & \Phi_{47} & \Phi_{48} \\ * & * & * & * & -\mathrm{e}^{-\alpha\tau}R_4 & 0 & 0 & 0 \\ * & * & * & * & * & -\dfrac{\alpha}{w_m^2 I} & 0 & 0 \\ * & * & * & * & * & * & -\varepsilon_1 I & 0 \\ * & * & * & * & * & * & * & -\varepsilon_2 I \end{bmatrix}, \tag{29}$$

$$\Phi_{11} = P_2^{\mathrm{T}}A_1 + A_1^{\mathrm{T}}P_2 + R_1 + R_2 - \mathrm{e}^{-\alpha h}S_1 + \alpha P_1 + hF_1 + hF_1^{\mathrm{T}} + hY_1 + hY_1^{\mathrm{T}} + \varepsilon_1 \lambda_1^2 I_n \quad (30)$$

$$\Phi_{12} = P_1 - P_2^{\mathrm{T}} + A_1^{\mathrm{T}}P_3 + hF_2 + hY_2 \quad (31)$$

$$\Phi_{13} = P_2^{\mathrm{T}}A_2 + Q_{12} - hF_1 + hF_3 + hX_1^{\mathrm{T}} + hY_3 \quad (32)$$

$$\Phi_{14} = \mathrm{e}^{-\alpha h}S_1 + hF_4 - hX_1^{\mathrm{T}} + hY_4 - hY_1^{\mathrm{T}} \quad (33)$$

$$\Phi_{15} = P_2^{\mathrm{T}}C + hF_5 + hY_5 \quad (34)$$

$$\Phi_{16} = P_2^{\mathrm{T}}B + hF_6 + hY_6 \quad (35)$$

$$\Phi_{17} = P_2^{\mathrm{T}}D_1 + hF_7 + hY_7 \quad (36)$$

$$\Phi_{18} = P_2^{\mathrm{T}}D_2 + hF_8 + hY_8 \quad (37)$$

$$\Phi_{22} = h^2(S_1 + S_2 + S_3) + h\mathrm{e}^{-\alpha h}Q_{11} + R_4 - P_3 - P_3^{\mathrm{T}} \quad (38)$$

$$\Phi_{23} = P_3^{\mathrm{T}}A_2 - hF_2^{\mathrm{T}} + hX_2^{\mathrm{T}} \quad (39)$$

$$\Phi_{33} = \quad hQ_{22} - Q_{12} - Q_{12}^{\mathrm{T}} - \min((1-h_d)\mathrm{e}^{-\alpha h}, 1 - h_d)R_1$$
$$+\max((1-h_d)\mathrm{e}^{-\alpha h}, 1 - h_d)R_3 \quad (40)$$
$$-hF_3 - hF_3^{\mathrm{T}} + hX_3 + hX_3^{\mathrm{T}} + \varepsilon_2 \lambda_2^2 I_n$$

$$\Phi_{34} = -hF_4 - hX_3^{\mathrm{T}} + hX_4 - hY_3^{\mathrm{T}} \quad (41)$$

$$\Phi_{35} = -hF_5 + hX_5^{\mathrm{T}} \quad (42)$$

$$\Phi_{36} = -hF_6 + hX_6^{\mathrm{T}} \quad (43)$$

$$\Phi_{37} = -hF_7 + hX_7^{\mathrm{T}}, \quad (44)$$

$$\Phi_{38} = -hF_8 + hX_8^{\mathrm{T}} \quad (45)$$

$$\Phi_{44} = -\mathrm{e}^{-\alpha h}[S_1 + R_2 + R_3] - hX_4 - hX_4^{\mathrm{T}} - hY_4 - hY_4^{\mathrm{T}} \quad (46)$$

$$\Phi_{45} = hX_5 - hX_5^{\mathrm{T}} \quad (47)$$

$$\Phi_{46} = hX_6 - hX_6^{\mathrm{T}} \quad (48)$$

$$\Phi_{47} = hX_7 - hX_7^{\mathrm{T}} \quad (49)$$

$$\Phi_{48} = hX_8 - hX_8^{\mathrm{T}} \quad (50)$$

Derived from Eq (26) through Eq (50), it follows that the ellipsoid $\Im(P_1, 1)$ defines the boundary of the RS for the NTSs.

Proof: To obtain a smaller RS boundary for the system Eq (2), we choose the following Lyapunov functional as

$$V(t) = V_1(t) + V_2(t) + V_3(t) + V_4(t) \quad (51)$$

where

$$V_1(t) = [\,x^{\mathrm{T}}(t) \quad \dot{x}^{\mathrm{T}}(t)\,] \begin{bmatrix} I & 0 \\ 0 & 0 \end{bmatrix} \begin{bmatrix} P_1 & 0 \\ P_2 & P_3 \end{bmatrix} \begin{bmatrix} x(t) \\ \dot{x}(t) \end{bmatrix} \tag{52}$$

$$
\begin{aligned}
V_2(t) = & \int_{t-h(t)}^{t} e^{\alpha(s-t)} x^{\mathrm{T}}(s) R_1 x(s) \mathrm{d}s + \int_{t-h}^{t} e^{\alpha(s-t)} x^{\mathrm{T}}(s) R_2 x(s) \mathrm{d}s \\
& + \int_{t-h}^{t-h(t)} e^{\alpha(s-t)} x^{\mathrm{T}}(s) R_3 x(s) \mathrm{d}s + \int_{t-\tau}^{t} e^{\alpha(s-t)} \dot{x}^{\mathrm{T}}(s) R_4 \dot{x}(s) \mathrm{d}s
\end{aligned}
\tag{53}
$$

$$V_3(t) = h \int_{-h}^{0} \mathrm{d}\theta \int_{t+\theta}^{t} e^{\alpha(s-t)} \dot{x}^{\mathrm{T}}(s)(S_1 + S_2 + S_3)\dot{x}(s)\mathrm{d}s \tag{54}$$

$$
\begin{aligned}
V_4(t) = & \int_{0}^{t} \mathrm{d}\theta \int_{\theta-h(\theta)}^{\theta} e^{\alpha(\theta-t)} [\,\dot{x}^{\mathrm{T}}(s) \quad x^{\mathrm{T}}(\theta-h(\theta))\,] \begin{bmatrix} Q_{11} & Q_{12} \\ Q_{12}^{\mathrm{T}} & Q_{22} \end{bmatrix} \begin{bmatrix} \dot{x}(s) \\ x(\theta-h(\theta)) \end{bmatrix} \mathrm{d}s \\
& + e^{\alpha h} \int_{0}^{t} \mathrm{d}\theta \int_{\theta-h(\theta)}^{\theta} e^{\alpha(s-t)} \dot{x}^{\mathrm{T}}(s) Q_{11} \dot{x}(s) \mathrm{d}s
\end{aligned}
\tag{55}
$$

The matrices $Q_{11}, Q_{22}, Q_{12}, P_1 > 0, P_2, P_3$, and $G > 0$, along with the matrices $R_1 > 0, R_2 > 0, R_3 > 0, S_1 \geqslant 0, S_2 > 0, S_3 > 0$, and the scalar $\alpha > 0$, are the solutions to the matrix inequalities specified in Eqs (26) to (28).

Firstly, it is straightforward to deduce that when $t - h \leqslant s \leqslant t$, we have

$$0 < e^{-\alpha h} \leqslant e^{\alpha(s-t)} \leqslant 1 \tag{56}$$

$$0 \leqslant h - t + s \leqslant h \tag{57}$$

hold true. Therefore, it can conclude:

$$V_3(t) \geqslant 0. \tag{58}$$

Further, we can deduce:

$$V_2(t) + V_3(t) + V_4(t) \geqslant 0 \tag{59}$$

Thus, we can derive:

$$V(t) = \sum_{i=1}^{5} V_i \geqslant V_1 x(t) = x^{\mathrm{T}}(t) P_1 x(t). \tag{60}$$

Next, we compute the time derivative of the Lyapunov functional $V(t)$ along the system's trajectory. This involves calculating how $V(t)$ changes over time as the system evolves.

$$V(t) = \dot{V}_1(t) + \dot{V}_2(t) + \dot{V}_3(t) + \dot{V}_4(t) + \dot{V}_5(t) \tag{61}$$

Through Eq (51), we get that

$$
\dot{V}_1(t) = \begin{bmatrix} x^{\mathrm{T}}(t) & \dot{x}^{\mathrm{T}}(t) \end{bmatrix} \begin{bmatrix} P_1 & P_2^{\mathrm{T}} \\ 0 & P_3^{\mathrm{T}} \end{bmatrix} \begin{bmatrix} x(t) \\ 0 \end{bmatrix}
$$

$$
= \begin{bmatrix} x^{\mathrm{T}}(t) & \dot{x}^{\mathrm{T}}(t) \end{bmatrix} \begin{bmatrix} P_1 & P_2^{\mathrm{T}} \\ 0 & P_3^{\mathrm{T}} \end{bmatrix} \begin{bmatrix} \dot{x}(t) \\ (A_1 x(t) - \dot{x}(t) + A_2 x(t - h(t)) + C\dot{x}(t - d(t)) \\ + Bw(t) + D_1 f_1(t, x(t)) + D_2 f_2(t, x(t - h(t))) \end{bmatrix}
$$

$$
= x^{\mathrm{T}}(t)[P_2^{\mathrm{T}} A_1 + A_1^{\mathrm{T}} P_2] x(t) + 2x^{\mathrm{T}}(t)[P_1 - P_2^{\mathrm{T}} + A_1^{\mathrm{T}} P_3]\dot{x}(t)
$$

$$
+ 2x^{\mathrm{T}}(t)P_2^{\mathrm{T}} A_2 x(t - h(t)) + 2x^{\mathrm{T}}(t)P_2^{\mathrm{T}} C\dot{x}(t - \tau) + 2x^{\mathrm{T}}(t)P_2^{\mathrm{T}} Bw(t)
$$

$$
+ 2x^{\mathrm{T}}(t)P_2^{\mathrm{T}} D_1 f_1(t, x(t)) + 2x^{\mathrm{T}}(t)P_2^{\mathrm{T}} D_2 f_2(t, x(t - h(t)))
$$

$$
- \dot{x}^{\mathrm{T}}(t)[P_3 + P_3^{\mathrm{T}}]\dot{x}(t) + 2\dot{x}^{\mathrm{T}}(t)P_3^{\mathrm{T}} A_2 x(t - h(t)) + 2x^{\mathrm{T}}(t)P_3^{\mathrm{T}} Bw(t) + 2\dot{x}^{\mathrm{T}}(t)P_3^{\mathrm{T}} C\dot{x}(t - \tau)
$$

$$
+ 2\dot{x}^{\mathrm{T}}(t)P_3^{\mathrm{T}} D_1 f_1(t, x(t)) + 2\dot{x}^{\mathrm{T}}(t)P_3^{\mathrm{T}} D_2 f_2(t, x(t - h(t))),
$$

(62)

$$
\dot{V}_2(t) \leqslant x^{\mathrm{T}}(t)[R_1 + R_2] x(t) + \dot{x}^{\mathrm{T}}(t)R_4 \dot{x}(t)
$$

$$
- \min((1 - h_d)\mathrm{e}^{-\alpha h}, 1 - h_d) \cdot x^{\mathrm{T}}(t - h(t))R_1 x(t - h(t))
$$

$$
+ \max((1 - h_d)\mathrm{e}^{-\alpha h}, 1 - h_d)x^{\mathrm{T}}(t - h(t)) \cdot R_3 x(t - h(t))
$$

$$
- \mathrm{e}^{-\alpha h} x^{\mathrm{T}}(t - h)[R_2 + R_3] x(t - h) - \mathrm{e}^{-\alpha \tau} \dot{x}^{\mathrm{T}}(t - \tau)R_4 \cdot \dot{x}(t - \tau) - \alpha V_2
$$

(63)

$$
\dot{V}_3(t) = h^2 \dot{x}^{\mathrm{T}}(t)[S_1 + S_2 + S_3]\dot{x}(t) - h \int_{t-h}^{t} \mathrm{e}^{\alpha(s-t)} \dot{x}^{\mathrm{T}}(s)S_1 \dot{x}(s)\mathrm{d}s
$$

$$
- h \int_{t-h}^{t} \mathrm{e}^{\alpha(s-t)} \dot{x}^{\mathrm{T}}(s)[S_2 + S_3]\dot{x}(s)\mathrm{d}s - \alpha V_3
$$

$$
\leqslant h^2 \dot{x}^{\mathrm{T}}(t)[S_1 + S_2 + S_3]\dot{x}(t) - \mathrm{e}^{-\alpha h}\left( \int_{t-h}^{t} \dot{x}(s)\mathrm{d}s \right)^{\mathrm{T}} S_1 \left( \int_{t-h}^{t} \dot{x}(s)\mathrm{d}s \right)
$$

(64)

$$
- h\mathrm{e}^{-\alpha h} \int_{t-h}^{t} \dot{x}^{\mathrm{T}}(s)[S_2 + S_3]\dot{x}(s)\mathrm{d}s - \alpha V_3
$$

$$
\leqslant - \mathrm{e}^{-\alpha h} x^{\mathrm{T}}(t)S_1 x(t) + h^2 \dot{x}^{\mathrm{T}}(t)[S_1 + S_2 + S_3]\dot{x}(t) + 2\mathrm{e}^{-\alpha h} x^{\mathrm{T}}(t)S_1
$$

$$\dot{V}_4(t) = \int_{t-h(t)}^t \dot{x}^\mathrm{T}(s)Q_{11}\dot{x}(s)\mathrm{d}s + 2\left(\int_{t-h(t)}^t \dot{x}^\mathrm{T}(s)\mathrm{d}s\right)^\mathrm{T} Q_{12}x(t-h(t))$$

$$+h(t)x^\mathrm{T}(t-h(t))Q_{22}x(t-h(t)) + h_M \mathrm{e}^{\alpha h}\dot{x}^\mathrm{T}(s)Q_{11}\dot{x}(s)$$

$$-\int_{t-h}^t \dot{x}^\mathrm{T}(s)Q_{11}\dot{x}(s)\mathrm{d}s - \alpha V_4$$

$$\leqslant \int_{t-h(t)}^t \dot{x}^\mathrm{T}(s)Q_{11}\dot{x}(s)\mathrm{d}s + 2[x(t)-x(t-h(t))]^\mathrm{T}Q_{12}x(t-h(t)) \tag{65}$$

$$+hx^\mathrm{T}(t-h(t))Q_{22}x(t-h(t)) + h\mathrm{e}^{\alpha h}\dot{x}^\mathrm{T}(s)Q_{11}\dot{x}(s) - \int_{t-h}^t \dot{x}^\mathrm{T}(s)Q_{11}\dot{x}(s)\mathrm{d}s - \alpha V_4$$

$$= 2x^\mathrm{T}(t)Q_{12}x(t-h(t)) + h\mathrm{e}^{\alpha h}\dot{x}^\mathrm{T}(s)Q_{11}\dot{x}(s)$$

$$+x^\mathrm{T}(t-h(t))[hQ_{22}-Q_{12}-Q_{12}^\mathrm{T}]x(t-h(t)) - \alpha V_4.$$

Then, we can futher get

$$\dot{V}(t) + \alpha V(t) - \frac{\alpha}{w_m^2}w^\mathrm{T}(t)w(t)$$

$$\leqslant x^\mathrm{T}(t)[P_2^\mathrm{T}A_1 + A_1^\mathrm{T}P_2 + R_1 + R_2 - \mathrm{e}^{-\alpha h}S_1 + \alpha P]x(t) + 2x^\mathrm{T}(t)[P_1 - P_2^\mathrm{T} + A_1^\mathrm{T}P_3]\dot{x}(t)$$

$$+2x^\mathrm{T}(t)[P_2^\mathrm{T}A_2 + Q_{12}]\cdot x(t-h(t)) + 2x^\mathrm{T}(t)\mathrm{e}^{-\alpha h}S_1 x(t-h) + 2x^\mathrm{T}(t)P_2^\mathrm{T}C\dot{x}(t-\tau)$$

$$+2x^\mathrm{T}(t)P_2^\mathrm{T}B\cdot w(t) + 2x^\mathrm{T}(t)P_2^\mathrm{T}D_1 f_1(t,x(t)) + 2x^\mathrm{T}(t)P_2^\mathrm{T}D_2 f_2(t,x(t-h(t)))$$

$$+\dot{x}^\mathrm{T}(t)\cdot [h^2(S_1+S_2+S_3) + h\mathrm{e}^{-\alpha h}Q_{11} + R_4 - P_3 - P_3^\mathrm{T}]\dot{x}(t)$$

$$+2\dot{x}^\mathrm{T}(t)P_3^\mathrm{T}A_2 x(t-h(t)) + 2\dot{x}^\mathrm{T}(t)P_3^\mathrm{T}C\dot{x}(t-\tau) + 2\dot{x}^\mathrm{T}(t)P_3^\mathrm{T}Bw(t)$$

$$+2\dot{x}^\mathrm{T}(t)P_3^\mathrm{T}D_1 f_1(t,x(t)) + 2x^\mathrm{T}(t)P_2^\mathrm{T}\cdot D_2 f_2(t,x(t-h(t))) \tag{66}$$

$$+x^\mathrm{T}(t-h(t))[hQ_{22}-Q_{12}-Q_{12}^\mathrm{T} - \min((1-h_d)\mathrm{e}^{-\alpha h}, 1-h_d)R_1$$

$$+\max((1-h_d)\mathrm{e}^{-\alpha h}, 1-h_d)R_3]x(t-h(t)) - h\mathrm{e}^{-\alpha h}\int_{t-h}^{t-h(t)} \dot{x}^\mathrm{T}(s)S_2\dot{x}(s)\mathrm{d}s$$

$$-\mathrm{e}^{-\alpha h}x^\mathrm{T}(t-h)[R_2+R_3]x(t-h) - \mathrm{e}^{-\alpha\tau}\dot{x}(t-\tau)R_4\dot{x}(t-\tau)$$

$$-h\mathrm{e}^{-\alpha h}\int_{t-h(t)}^t \dot{x}^\mathrm{T}(s)S_2\dot{x}(s)\mathrm{d}s - h\mathrm{e}^{-\alpha h}\int_{t-h}^t \dot{x}^\mathrm{T}(s)S_3\dot{x}(s)\mathrm{d}s - \frac{\alpha}{w_m^2}w^\mathrm{T}(t)w(t)$$

$$\leqslant \xi^\mathrm{T}(t)\left[\hat{\Phi} + h^2\mathrm{e}^{\alpha h}F^\mathrm{T}S_1^{-1}F + h^2\mathrm{e}^{\alpha h}X^\mathrm{T}S_2^{-1}X + h^2\mathrm{e}^{\alpha h}Y^\mathrm{T}S_3^{-1}Y\right]\xi(t)$$

where

$$\hat{\Phi} = \begin{bmatrix} \hat{\Phi}_{11} & \Phi_{12} & \Phi_{13} & \Phi_{14} & \Phi_{15} & \Phi_{16} & \Phi_{17} & \Phi_{18} \\ * & \Phi_{22} & \Phi_{23} & -hX_2^{\mathrm{T}} - hY_2^{\mathrm{T}} & P_3^{\mathrm{T}}C & P_3^{\mathrm{T}}B & P_3^{\mathrm{T}}D_1 & P_3^{\mathrm{T}}D_2 \\ * & * & \hat{\Phi}_{33} & \Phi_{34} & \Phi_{35} & \Phi_{36} & \Phi_{37} & \Phi_{38} \\ * & * & * & \Phi_{44} & \Phi_{45} & \Phi_{46} & \Phi_{47} & \Phi_{48} \\ * & * & * & * & -e^{-\alpha\tau}R_4 & 0 & 0 & 0 \\ * & * & * & * & * & -\dfrac{\alpha}{w_m^2 I} & 0 & 0 \\ * & * & * & * & * & * & 0 & 0 \\ * & * & * & * & * & * & * & 0 \end{bmatrix} \tag{67}$$

$$\Phi_{11} = P_2^{\mathrm{T}}A_1 + A_1^{\mathrm{T}}P_2 + R_1 + R_2 - e^{-\alpha h}S_1 + \alpha P_1 + hF_1 + hF_1^{\mathrm{T}} + hY_1 + hY_1^{\mathrm{T}}, \tag{68}$$

$$\begin{aligned} \Phi_{33} = \quad & hQ_{22} - Q_{12} - Q_{12}^{\mathrm{T}} - \min((1-h_d)e^{-\alpha h}, 1-h_d)R_1 \\ & + \max((1-h_d)e^{-\alpha h}, 1-h_d)R_3 - hF_3 - hF_3^{\mathrm{T}} + hX_3 + hX_3^{\mathrm{T}}. \end{aligned} \tag{69}$$

Based on condition Eqs (6) to (7) and using the *S*-procedure, we can derive the following inequality:

$$\begin{aligned} & \dot{V}(t) + \alpha V(t) - \frac{\alpha}{w_m^2} w^{\mathrm{T}}(t)w(t) \\ \leqslant \quad & \xi^{\mathrm{T}}(t)\left[\hat{\Phi} + h^2 e^{\alpha h}F^{\mathrm{T}}S_1^{-1}F + h^2 e^{\alpha h}X^{\mathrm{T}}S_2^{-1}X + h^2 e^{\alpha h}Y^{\mathrm{T}}S_3^{-1}Y\right]\xi(t) \\ & + \varepsilon_1[\lambda_1^2 x^{\mathrm{T}}(t)x(t) - f_1^{\mathrm{T}}(x, x(t))f_1(x, x(t))f_1(x, x(t))] \\ & \varepsilon_2[\lambda_2^2 x^{\mathrm{T}}(t-h(t))x(t-h(t)) - f_2^{\mathrm{T}}(x, x(t-h(t)))f_2(x, x(t-h(t)))]. \end{aligned} \tag{70}$$

From the matrix inequalities Eqs (26) to (27), we can derive:

$$\begin{aligned} & \dot{V}(t) + \alpha V(t) - \frac{\alpha}{w_m^2} w^{\mathrm{T}}(t)w(t) \\ \leqslant \quad & \xi^{\mathrm{T}}(t)\left[\hat{\Phi} + h^2 e^{\alpha h}F^{\mathrm{T}}S_1^{-1}F + h^2 e^{\alpha h}X^{\mathrm{T}}S_2^{-1}X + h^2 e^{\alpha h}Y^{\mathrm{T}}S_3^{-1}Y\right]\xi(t) \\ & + \varepsilon_1[\lambda_1^2 x^{\mathrm{T}}(t)x(t) - f_1^{\mathrm{T}}(x, x(t))f_1(x, x(t))] \\ & + \varepsilon_2[\lambda_2^2 x^{\mathrm{T}}(t-h(t))x(t-h(t)) - f_2^{\mathrm{T}}(x, x(t-h(t)))f_2(x, x(t-h(t)))] \leqslant 0. \end{aligned} \tag{71}$$

Thus, we can conclude:

$$\dot{V}(t) + \alpha V(t) - \frac{\alpha}{w_m^2} w^{\mathrm{T}}(t)w(t) \leqslant 0. \tag{72}$$

Based on Lemma 3, we can determine

$$V(t) = V_1(t) + V_2(t) + V_3(t) + V_4(t) \leqslant 1 \tag{73}$$

Furthermore, inequality Eq (51) implies $V_2(t) + V_3(t) + V_4(t) > 0$, hence:

$$V_1(t) = x^{\mathrm{T}}(t) P_1 x(t) \leqslant 1 \tag{74}$$

is valid. With this, the theorem is proven.

**Remark 3:** Solving the matrix inequalities in Theorem 1 does not ensure their uniqueness. This is attributed to the fact that minimizing $\mathrm{logdet}(P_1)^{1/2}$ also minimizes the volume of the associated ellipsoid $\mathfrak{J}(P_1, 1)$. We generally use $\mathrm{logdet}(P_1)^{1/2}$ to estimate the volume of the ellipsoid $\mathfrak{J}(P_1, 1)$. Nonetheless, determining the minimum of $\mathrm{logdet}(P_1)^{1/2}$ can be exceedingly challenging. Consequently, the procedure is often simplified to finding the largest $\delta$ for which $\delta I \leqslant P_1$, as follows:

$$\min \bar{\delta} \left( \bar{\delta} = \frac{1}{\delta} \right)$$

$$\text{s.t.} \begin{cases} (a) \begin{bmatrix} \bar{\delta} I & I \\ I & P_1 \end{bmatrix} \geqslant 0, \\[4mm] (b) \quad P_1 > 0 \text{ and satisfy matrix inequality } Eq.(29). \end{cases} \tag{75}$$

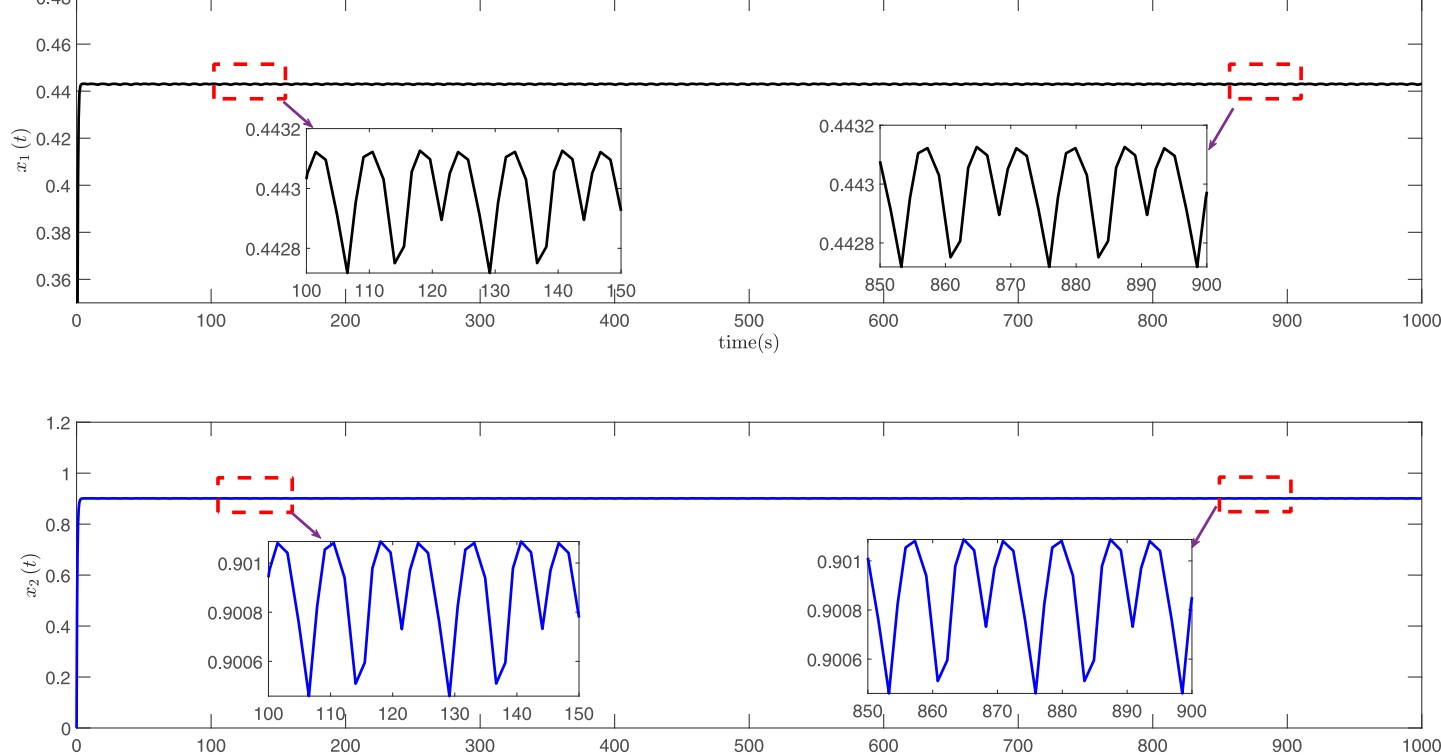

**Fig 1. The response diagram of system state $x(t)$ under constant disturbance.**

## 4 Numerical simulation verification

In this simulation, the NTS analyzed in this section is characterized as follows:

$$\dot{x}(t) = A_1 x(t) + A_2 x(t - h(t)) + Bw(t) + D_1 f_1(t, x(t)) + D_2 f_2(t, x(t - h(t))) \tag{76}$$

where

$$A_1 = \begin{bmatrix} -1.2 & 0 \\ 0 & -0.55 \end{bmatrix}, \quad A_2 = \begin{bmatrix} -0.6 & 0.7 \\ -1 & -0.8 \end{bmatrix}, \quad B = \begin{bmatrix} 0.6 \\ 0.8 \end{bmatrix} \quad C = \begin{bmatrix} 0 & 0 \\ 0 & 0 \end{bmatrix}$$

$$D_1 = \begin{bmatrix} -0.1 & 0 \\ 0 & -0.1 \end{bmatrix}, \quad D_2 = \begin{bmatrix} -0.1 & 0 \\ 0 & -0.1 \end{bmatrix}, \quad w^T(t)w(t) < w_m^2 = 1$$

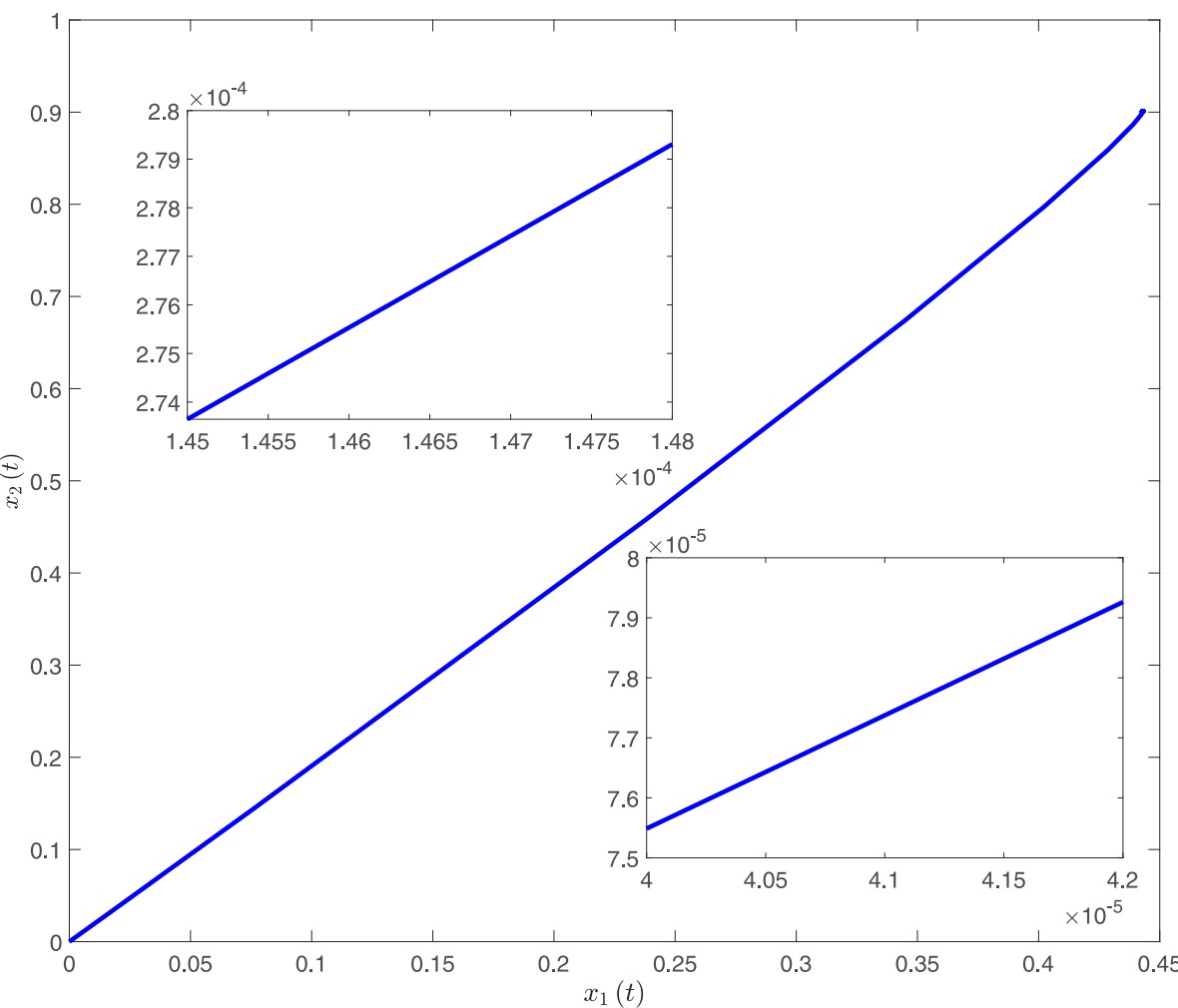

**Fig 2. The response diagram of system state trajectory under constant disturbance.**

and $f_1(t, x(t))$ and $f_2(t, x(t - h(t)))$ satisfy the following inequality

$$f_1^{\mathrm{T}}(t, x(t))f_1(t, x(t)) \leqslant 0.01x^{\mathrm{T}}(t)x(t) \tag{77}$$

$$f_2^{\mathrm{T}}(t, x(t - h(t)))f_2(t, x(t - h(t))) \leqslant 0.05x^{\mathrm{T}}(t - h(t))x(t - h(t)) \tag{78}$$

In the control system, the disturbance of $w(t)$ will cause the change of system state, which may lead to the decrease of stability, oscillation and even instability. On the other hand, the time-varying disturbance $w(t)$ will make the behavior of the system more complicated, because the disturbance will change with time. In order to verify the proposed theory and observe the corresponding reaction, two simulation scenarios with different disturbance situations are designed in this section. Specifically, we set two different disturbance scenarios: one is constant disturbance, which is set to $w(t) = 0.55$. The other is time-varying disturbance, which is set to $w(t) = 0.55 + 0.2\,sin(0.2t)$ to observe the dynamic response of the system under these two different disturbance conditions.

This paper presents simulation results in Figs 1 through 6. Fig 1 illustrates the time response of the state vector for NTSs subjected to a constant disturbance. Fig 2 depicts the state trajectory of the system with a constant disturbance and a time delay of $h = 0.70$. These figures demonstrate that the system's state remains stable within a specific range when exposed to constant disturbances. Furthermore, Fig 3 contrasts the state trajectories with a constant disturbance of $w(t) = 0.55$ against ellipsoidal RS boundaries based on Theorem 1 from this study, as well as those established using the Kim method (see Reference [39]) and the Zuo method (see Reference [40]). The ellipsoidal boundaries for the RS generated by the method described in Theorem 1 are notably smaller than those determined by the prior methods. Figs 4 and 5

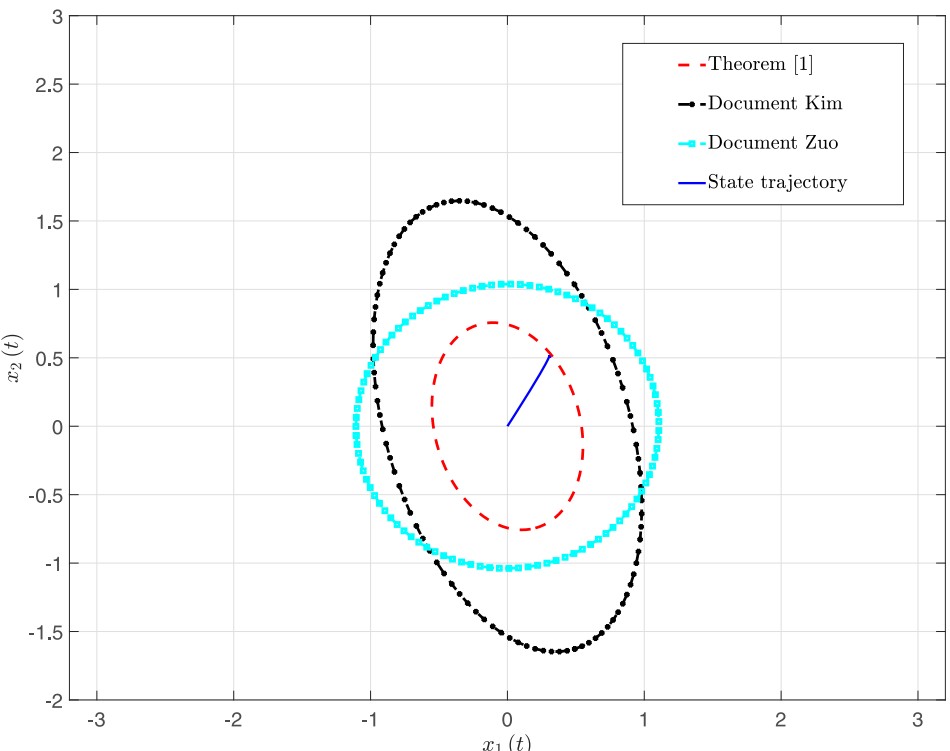

**Fig 3. The RS system state trajectory diagram of different methods under constant disturbance.**

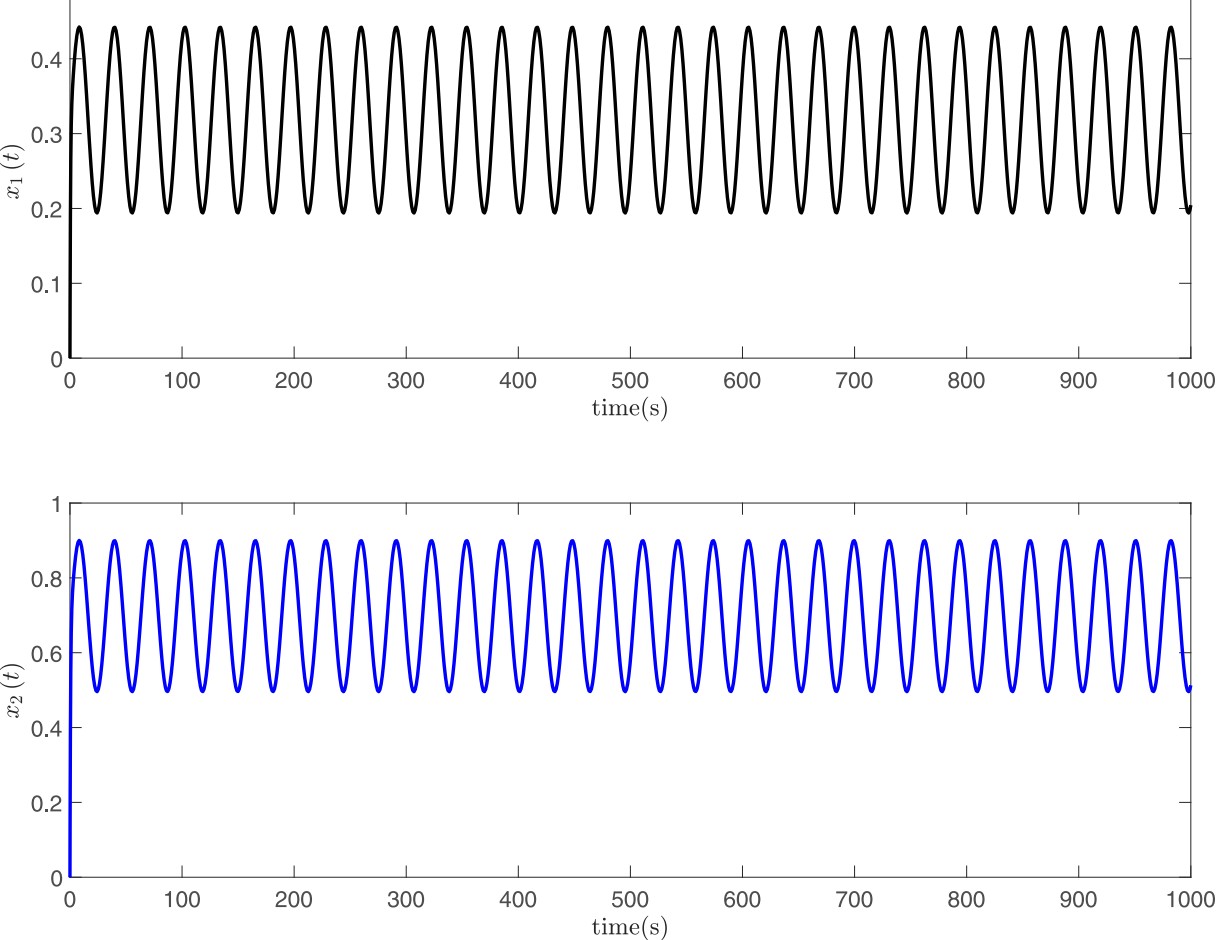

**Fig 4. The response graph of the system state $x(t)$ subject to time-varying disturbance.**

present the state quantity and trajectory diagrams under a time-varying disturbance, indicating that the system's state and trajectory remain relatively stable under the conditions set by the proposed method. Finally, Fig 6 compares the RS of the nonlinear NTSs using the proposed method against the methods referenced in [39, 40]. This comparison shows that the proposed method not only ensures complete containment of the system state trajectory but also achieves a smaller RS compared to existing methods.

## 5 Discussion and future work

This study tackles the problem of establishing RS ellipsoidal boundaries for neutral type time-delay systems with nonlinear and bounded disturbances. We crafted a Lyapunov function and leveraged matrix inequality techniques, along with free weighting matrices, to precisely delineate the RS ellipsoidal boundaries. Our numerical examples indicate that this method provides tighter and more accurate boundaries than those found in existing literature. Additionally, we identify machine learning as a promising approach for addressing RS challenges in complex, high-dimensional systems. Future work will focus on incorporating machine learning strategies to further refine and optimize RS boundary estimations.

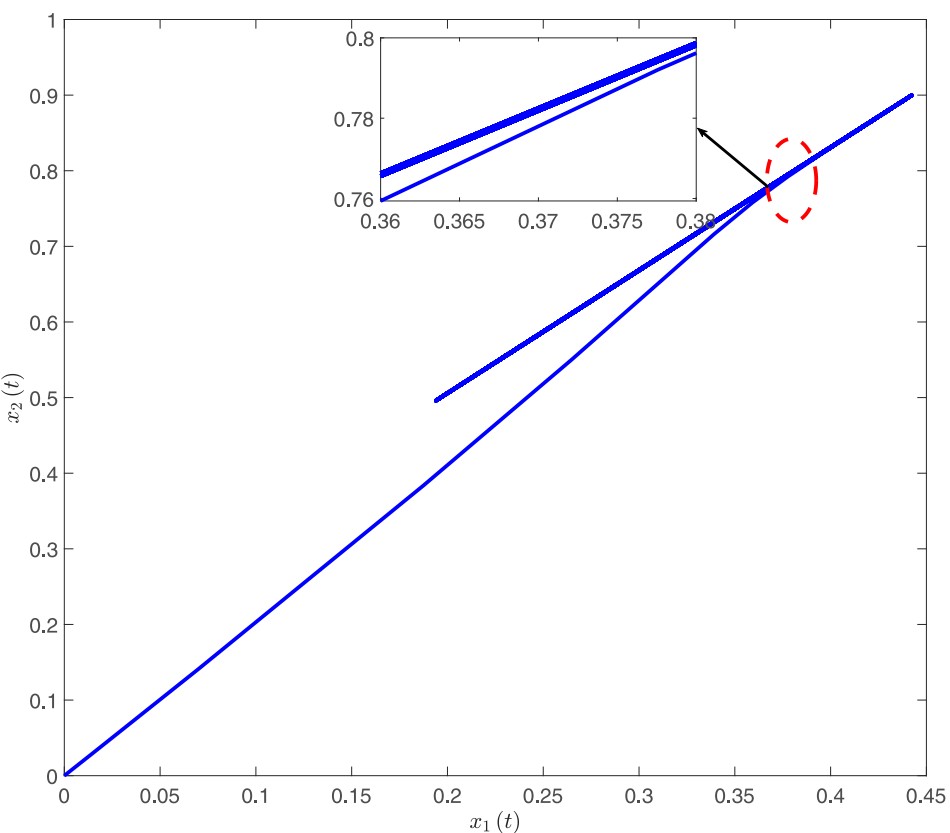

**Fig 5. The response diagram of system state trajectory subject to time-varying disturbance.**

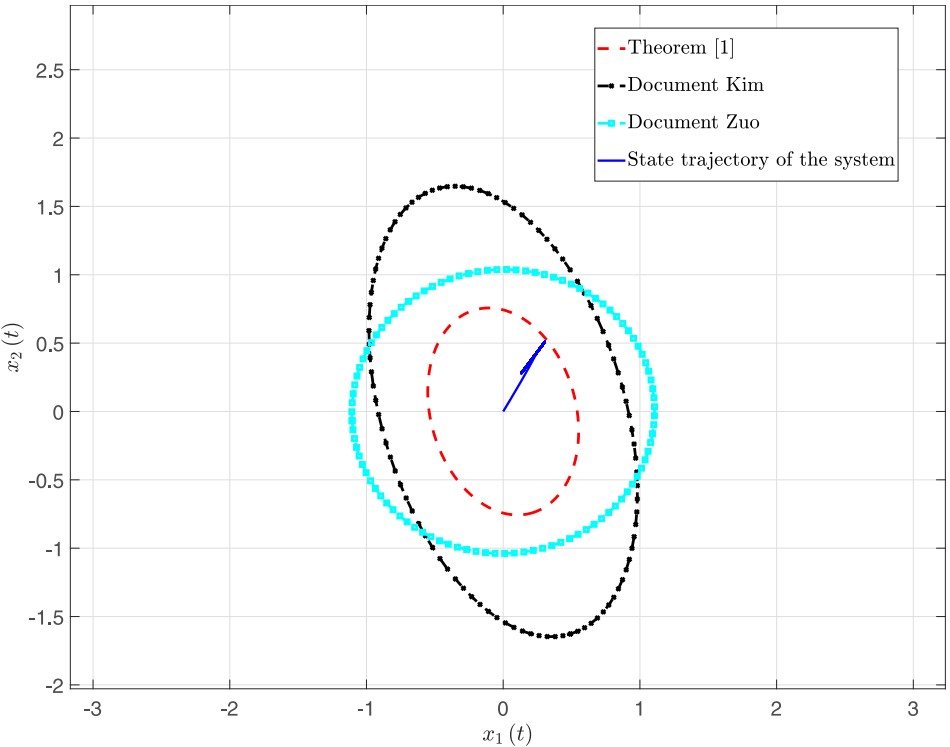

**Fig 6. The trajectory diagrams of the RS system state for various methods under time-varying disturbance.**

## Supporting information

**S1 File.**
(PDF)

## Author Contributions

**Data curation:** Dongmei Xia, Kaiyuan Chen.

**Methodology:** Dongmei Xia, Kaiyuan Chen, Lin Sun.

**Writing – original draft:** Dongmei Xia, Kaiyuan Chen.

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
