## [Decision Letter · Decision Letter 0]

9 Dec 2024

PONE-D-24-52895Research on reachable set boundary of neutral system with various types of disturbancesPLOS ONE

Dear Dr. Xia,

Thank you for submitting your manuscript to PLOS ONE. After careful consideration, we feel that it has merit but does not fully meet PLOS ONE’s publication criteria as it currently stands. Therefore, we invite you to submit a revised version of the manuscript that addresses the points raised during the review process.

We look forward to receiving your revised manuscript.

Kind regards,

Dr. Guojin Qin

Academic Editor

PLOS ONE

Journal Requirements:

This work is partially supported by the Guangdong Province Ordinary Universities Key Areas Special Project (High-end Equipment Manufacturing), Project Number: 2022ZDZX3074. 

5. We are unable to open your Supporting Information file S1.zip. Please kindly revise as necessary and re-upload.

Reviewers' comments:

Reviewer's Responses to Questions

**Comments to the Author**

1. Is the manuscript technically sound, and do the data support the conclusions?

Reviewer #1: Yes

Reviewer #2: Yes

2. Has the statistical analysis been performed appropriately and rigorously? 

Reviewer #1: Yes

Reviewer #2: Yes

3. Have the authors made all data underlying the findings in their manuscript fully available?

Reviewer #1: Yes

Reviewer #2: Yes

4. Is the manuscript presented in an intelligible fashion and written in standard English?

Reviewer #1: Yes

Reviewer #2: Yes

5. Review Comments to the Author

Reviewer #1: This paper is devoted to reachable set boundary of neutral system with various types of disturbances. Some comments are given below:

1. Please enrich your Introduction with extended disturbance-observer-based data-driven control of networked nonlinear systems, output anti-disturbance control of stochastic Markov jump systems with multiple disturbances.

2. If Lemmas 6 and 7 are borrowed from existing results, please give the corresponding citation. Otherwise, please supply the full proof details.

3.Please make some necessary comparisons with existing results on theory with some comments.

Reviewer #2: In this paper, the reachable set for NTSs is studied by using the Lyapunov functional conjunction with various matrix inequality techniques. And the safe, efficient, and minimized boundaries of the RS for NTSs is demonstrated via an example. In addition, there are quite a few problems in the paper. Specific suggestions for improvement of the manuscript:

1. In the title, various types of disturbances are shown, but how to dealt with them is not explained in the paper?

2. In the article, the system includes the problem of time delay, but The issue of time delay was not considered in the abstract.

3. The abstract is not an introduction, please revise the abstract carefully. There are too many things in the introduction that are not relevant to the content of this article. There is almost no background to this paper, just a mention of the goals that the proposed model needs to achieve. Moreover, it provides too little introduction to relevant studies on the detection methods. Some ideas like Integral reinforcement learning-based dynamic event-triggered safety control for multiplayer Stackelberg–Nash games with time-varying state constraints, Engineering Applications of Artificial Intelligence; Parallel learning-based security robust tracking control for nonlinear systems with uncertainties: An event-triggered design, , Engineering Applications of Artificial Intelligence; Barrier-critic adaptive robust control of nonzero-sum differential games for uncertain nonlinear systems with state constraints,IEEE Transactions on Systems, Man, and Cybernetics: Systems. can obviously enhance the article's quality and impact.

4. In addition, the main contributions described in this article are vague. The main contributions described in this article does not describe how to handle disturbances.

5. How to get the equation (2)?

6. In the equation (2), τ (t) is a time-varying delay?

7. In Lyapunov functional (51), Where is the innovation reflected in the equation (51)?

6. PLOS authors have the option to publish the peer review history of their article (what does this mean?). If published, this will include your full peer review and any attached files.

Reviewer #1: No

Reviewer #2: No

---

## [Decision Letter · Decision Letter 1]

29 Dec 2024

Research on reachable set boundary of neutral system with various types of disturbances

PONE-D-24-52895R1

Dear Dr. Xai,

We’re pleased to inform you that your manuscript has been judged scientifically suitable for publication and will be formally accepted for publication once it meets all outstanding technical requirements.

Kind regards,

Dr. Guojin Qin

Academic Editor

PLOS ONE

Additional Editor Comments (optional):

Reviewers' comments:

Reviewer's Responses to Questions

**Comments to the Author**

1. If the authors have adequately addressed your comments raised in a previous round of review and you feel that this manuscript is now acceptable for publication, you may indicate that here to bypass the “Comments to the Author” section, enter your conflict of interest statement in the “Confidential to Editor” section, and submit your "Accept" recommendation.

Reviewer #1: All comments have been addressed

Reviewer #2: All comments have been addressed

2. Is the manuscript technically sound, and do the data support the conclusions?

Reviewer #1: Yes

Reviewer #2: Yes

3. Has the statistical analysis been performed appropriately and rigorously? 

Reviewer #1: Yes

Reviewer #2: Yes

4. Have the authors made all data underlying the findings in their manuscript fully available?

Reviewer #1: (No Response)

Reviewer #2: Yes

5. Is the manuscript presented in an intelligible fashion and written in standard English?

Reviewer #1: (No Response)

Reviewer #2: Yes

6. Review Comments to the Author

Reviewer #1: (No Response)

Reviewer #2: The authors have revisited manuscript carefully. And the theory is reasonable and has application value. Therefore, there are no more comments, it could be accepted at present form.

7. PLOS authors have the option to publish the peer review history of their article (what does this mean?). If published, this will include your full peer review and any attached files.

Reviewer #1: No

Reviewer #2: No

---

## [Editor Report · Acceptance letter]

5 Jan 2025

PONE-D-24-52895R1 

PLOS ONE

Dear Dr. Xia, 

I'm pleased to inform you that your manuscript has been deemed suitable for publication in PLOS ONE. Congratulations! Your manuscript is now being handed over to our production team.

Kind regards, 

on behalf of

Dr. Guojin Qin 

Academic Editor

PLOS ONE